# Immunogenicity Following Two Doses of the BBIBP-CorV Vaccine and a Third Booster Dose with a Viral Vector and mRNA COVID-19 Vaccines against Delta and Omicron Variants in Prime Immunized Adults with Two Doses of the BBIBP-CorV Vaccine

**DOI:** 10.3390/vaccines10071071

**Published:** 2022-07-03

**Authors:** Jira Chansaenroj, Nungruthai Suntronwong, Sitthichai Kanokudom, Suvichada Assawakosri, Ritthideach Yorsaeng, Preeyaporn Vichaiwattana, Sirapa Klinfueng, Lakana Wongsrisang, Donchida Srimuan, Thaksaporn Thatsanatorn, Thanunrat Thongmee, Chompoonut Auphimai, Pornjarim Nilyanimit, Nasamon Wanlapakorn, Natthinee Sudhinaraset, Yong Poovorawan

**Affiliations:** 1Center of Excellence in Clinical Virology, Department of Pediatrics, Faculty of Medicine, Chulalongkorn University, Bangkok 10330, Thailand; job151@hotmail.com (J.C.); suntronwong.n@gmail.com (N.S.); kanokudom_s@yahoo.com (S.K.); suvichada.assawa@gmail.com (S.A.); ritthideach.yor@gmail.com (R.Y.); preeya_teiy@hotmail.com (P.V.); sirapa.klinfueng@gmail.com (S.K.); lakkhana4118@gmail.com (L.W.); donchida.s@gmail.com (D.S.); thaksapohnl@hotmail.com (T.T.); tata033@hotmail.com (T.T.); chompoonut.bit@gmail.com (C.A.); mim_bhni@hotmail.com (P.N.); nasamon.w@chula.ac.th (N.W.); dr_natthinee@hotmail.com (N.S.); 2Osteoarthritis and Musculoskeleton Research Unit, Department of Biochemistry, Faculty of Medicine, Chulalongkorn University, Bangkok 10330, Thailand; 3Division of Academic Affairs, Faculty of Medicine, Chulalongkorn University, Bangkok 10330, Thailand; 4The Royal Society of Thailand (FRS(T)), Sanam Sueapa, Dusit, Bangkok 10330, Thailand

**Keywords:** third dose, inactivated vaccine, BBIBP-CorV, viral vector vaccine, mRNA vaccine

## Abstract

Coronavirus disease 2019 (COVID-19) booster vaccination is being comprehensively evaluated globally due to waning immunity and the emergence of new severe acute respiratory syndrome coronavirus 2 (SARS-CoV-2) variants. Therefore, this study aimed to evaluate antibody responses in individuals vaccinated with two doses of the BBIBP-CorV vaccine and to explore the boosting effect of the different vaccine platforms in BBIBP-CorV-primed healthy adults, including a viral vector vaccine (AZD122) and mRNA vaccines (BNT162b2 and mRNA-1273). The results showed that in the BBIBP-CorV prime group, the total receptor-binding domain (RBD) immunoglobulin (Ig) and anti-RBD IgG levels waned significantly at three months after receiving the second dose. However, after the booster, RBD-specific binding antibody levels increased. Neutralizing antibody measured by a surrogate neutralization test showed inhibition over 90% against the SARS-CoV-2 delta variant but less than 70% against the omicron variant after the third dose on day 28. All booster vaccines could induce the total IFN-ɣ T-cell response. The reactogenicity was acceptable and well-tolerated without serious adverse events. This study supports the administration of the third dose with either a viral vector or mRNA vaccine for BBIBP-CorV-primed individuals to stimulate antibody and T-cell responses.

## 1. Introduction

The coronavirus disease 2019 (COVID-19) pandemic is caused by severe acute respiratory syndrome coronavirus 2 (SARS-CoV-2). It can result in mild to severe illness leading to hospitalization and death [1]. Currently available COVID-19 vaccines have been effective for the control and prevention of COVID-19 outbreaks. During this pandemic, many COVID-19 vaccines have been approved by the World Health Organization (WHO) for emergency use and are effective against SARS-CoV-2 in adults [2,3]. To date, inactivated vaccine candidates have been reported to be generally safe and have induced antibody responses in adults in preclinical trials [4]. The BBIBP-CorV vaccine is a killed inactivated coronavirus vaccine prepared by β-propionolactone [5]. In May 2021, the BBIBP-CorV vaccine was the first Chinese vaccine granted approval by the WHO for emergency use. Clinical trials on the BBIBP-CorV vaccine showed that BBIBP-CorV could protect against SARS-CoV-2 infection. The average quantitative IgG anti-spike protein antibodies’ levels among the participants on day 18 after vaccination with the BBIBP-CorV vaccine with no previous SARS-CoV-2 infection history was 40 AU/mL [6]. The most common reported adverse reactions of the BBIBP-CorV vaccine included mild pain at the injection site, fever, and fatigue [7]. Phase 3 clinical trials between July and December 2020 in seven countries have shown that two doses of the BBIBP-CorV vaccine had an efficacy of 78.1% against symptomatic cases, 100% protection against severe disease, and a 99% rate of seroconversion [4]. However, the antibody titers waned over time and may not be adequate to protect against the omicron variant [4,8]. Therefore, the Centre for Disease Prevention and Control China recommended a booster dose for BBIBP-CorV-primed individuals to stimulate adequate protection against the newly emerged omicron variant [9].

Although the efficacy of the vaccine and the ability to induce immune responses in two-dose-immunized participants can induce robust immunity, breakthrough infection with COVID-19 has been reported worldwide [2,3]. Several studies have observed that the decline in immunity may be related to a variant virus, weakening of vaccine-induced immunity, neutralizing antibody titers, or the effectiveness of the vaccine against symptomatic illness gradually weakening over time [2,3,10,11]. A study evaluating two doses of BBIBP-CorV vaccine effectiveness against SARS-CoV-2 variants showed efficacies of 39.2% for beta, 33.9% for delta, and 11.3% for alpha variants [12]. The emergence of the SARS-CoV-2 omicron variant (B.1.1.529), with several mutations recently identified in their spike, has become the predominant variant worldwide since November 2021 [13]. In the interim statement on 16 December 2021, the WHO recommended that viral vector or mRNA vaccines can be considered for third doses in those who received inactivated vaccines for the initial doses [14]. Studies from the United States have found an increased estimate of vaccine effectiveness against delta and omicron variants among adults who had received a third dose of an mRNA vaccine compared with that in those who were unvaccinated and those who received two doses [15,16].

The immunogenicity of COVID-19 booster vaccinations has been tested in a variety of situations with various vaccines. Both heterologous and homologous third booster shots have a significantly improved immune response [17,18,19]. A booster vaccination significantly increases neutralization capacity against SARS-CoV-2 variants, thus overcoming waning immunity and circulation of SARS-CoV-2 variants and significantly reducing the rates of confirmed infection and severe disease [20,21]. To improve immunity and enhance resistance to variants, a third dose of COVID-19 vaccination is necessary.

Antibody and T-cell responses against SARS-CoV-2 in two-dose BBIBP-CorV-primed individuals after the administration of the third dose of BNT162b2 have achieved a significant increase in both humoral and T-cell-mediated immune response [22]. The immunogenicity after a booster dose with a viral vector or other mRNA vaccine (mRNA1273) in BBIBP-CorV-primed individuals is limited. This study aimed to determine the kinetic response of antibodies in a cohort participant who completed the BBIBP-CorV vaccine and explore the immunogenicity of third doses with the BBIBP-CorV vaccine or a heterologous booster COVID-19 vaccination using viral vector and mRNA vaccines. The results could provide an opportunity to improve the flexibility and reliability of vaccination programs in the face of the COVID-19 pandemic.

## 2. Materials and Methods

### 2.1. Study Design and Population Study

This study was a prospective cohort study including two groups of participants. The first group consisted of individuals who consented to receive two doses of the BBIBP-CorV vaccine (referred to as the BBIBP-CorV-primed group). The second group comprised individuals primed with the two-dose BBIBP-CorV vaccine who were enrolled for the third booster (referred to as the third-booster group). The binding and neutralizing antibody titer and the T-cell response before and after the third dose of vaccination were evaluated. The inclusion criteria were immunocompetent individuals (age: ≥18 years) with no or well-controlled comorbidities and no previous SARS-CoV-2 infection in the medical history.

The BBIBP-CorV-primed group consisted of 51 participants who received the BBIBP-CorV vaccine at Synphaet Srinagarindra Hospital, Bangkok, Thailand between June and July 2021. Blood samples from this group were collected at four time points, day 0 (before receiving the first dose), day 30 ± 7 days (just before receiving the second dose), day 60 ± 7 days after receiving the first dose, and day 120 ± 7 after receiving the first dose.

The third-booster group consisted of 144 participants who received two prime doses of BBIBP-CorV over 6 ± 1 months and were enrolled to receive the third dose at the Center of Excellence in Clinical Virology, Department of Pediatrics, Faculty of Medicine, Chulalongkorn University. The participants were enrolled between November and December 2021. This group was further divided into three subgroups who were to receive the third booster dose (AZD1222, BNT162b2, or mRNA-1273). The third dose was administered according to vaccine availability. The participants and investigators were not blinded. The blood samples from this group were collected at four time points: before the third-dose vaccination (day 0, baseline) and after receiving the booster dose (day 14 ± 7 days, day 28 ± 7 days, and day 90 ± 7 days).

This study protocol was approved by the Institutional Review Board of the Faculty of Medicine of Chulalongkorn University (IRB numbers 192/64 and 546/64). Informed consent was obtained from all subjects involved in the study prior to enrollment in the study. All volunteers were informed about the details of the study prior to its start. The study was conducted in accordance with the Declaration of Helsinki and the Good Clinical Practice Guidelines (ICH-GCP).

### 2.2. Study Vaccines Interventions

BBIBP-CorV (Vero cell) (Sinopharm, National Biotec Group Co., Beijing Institute of Biological Products, Beijing, China) is an inactivated virus vaccine developed from whole SARS-CoV-2 strain HB02. The HB02 strain is obtained by passaging and purification in Vero cells. Consequently, the entire virion is inactivated in β-propiolactone and is further absorbed with aluminum hydroxide. One dose (0.5 mL) contains 6.50 U [5]. AZD1222 (ChAdOx1-S/nCoV-19, University of Oxford/AstraZeneca, Oxford, UK) is a nonreplicating chimpanzee adenovirus Oxford 1 vector vaccine presenting the SARS-CoV-2 spike protein. The virion is produced in genetically modified HEK293 cells. One dose (0.5 mL) contains 5 × 10^10^ infectious units [23]. BNT162b2 (Pfizer-BioNTech Inc., New York, NY, USA) and mRNA-1273 (ModernaTX Inc., Cambridge, MA, USA) are modified RNA containing lipid nanoparticles encoding the full-length spike of SARS-CoV-2, modified by two proline mutations to lock it in the prefusion conformation. One dose contains 30 and 100 µg for BNT162b2 and mRNA-1273, respectively [24,25]. After receiving the additional dose, all study participants were monitored for immunogenicity and safety.

### 2.3. Monitoring of Adverse Events

The participants were informed of the name of the vaccine and were observed for adverse events for at least 30 min after vaccination. All participants were continuously monitored for adverse events after immunization (AEFI) for seven days after receiving the booster dose using self-assessment records via online or paper questionnaires. The adverse events included system reactions and local reactions. System reactions included fever, headache, myalgia, nausea, vomiting, diarrhea, joint pain, chills, and dizziness. Local reactions included pain at the injection site, swelling, and redness.

### 2.4. Laboratory Experiments

#### 2.4.1. Immunoglobulin Assays

All serum samples were analyzed using SARS-CoV-2 total receptor binding domain immunoglobulin (total RBD Ig) and anti-RBD IgG by using the Electrochemiluminescence Immunoassay (ECLIA) (Roche Diagnostics GmbH, Mannheim, Germany) and the chemiluminescent microparticle immunoassay (CMIA) (Abbott Diagnostics, Sligo, Ireland), respectively [26]. A total RBD Ig level of ≥0.8 U/mL was considered positive. The anti-RBD IgG level with a value ≥7.1 BAU/mL was considered positive.

#### 2.4.2. Surrogate Virus Neutralization Test (sVNT)

To assess SARS-CoV-2 surrogate virus neutralization, a subset of serum samples from the third-booster group at all time points was also evaluated for neutralizing activity against the SARS-CoV-2 variants of concern (VOCs), namely B.1.617.2 (delta) and B.1.1.529 (omicron), using an ELISA-based surrogate virus neutralization test (sVNT) based on antibody-mediated blockage of the interaction between the viral receptor binding domain (RBD) and the angiotensin-converting enzyme 2 (ACE2) protein [27]. Furthermore, a cPass^TM^ SARS-CoV-2 neutralizing antibody detection kit (GenScript Biotech, Piscataway, NJ, USA) was used for all strains following the manufacturer’s instructions. Recombinant RBDs from B.1.617.2 (containing L452R and T478K) and B.1.1.529 (containing N501Y, E484A, K417N, and D614G) were also used with this kit. The ability of a serum to inhibit binding between RBD and ACE2 was calculated as a percentage as follows: 1 − (average OD of the sample/average OD of the negative control) × 100. A value ≥30% was considered positive, indicating the presence of neutralizing antibodies. The lower limit of detection was established as 0% inhibition.

#### 2.4.3. IFN-ɣ Releasing Assay

The subset of participants’ whole heparinized blood samples from the third-booster group was taken at all time points on days 0, 14, 28, and 90 to assess the specific T-cell response. The heparinized whole blood sample was collected in a blood collection tube from QuantiFERON (QFN) SARS-CoV-2 RUO kit (QIAGEN, Hilden, Germany) following the manufacturer’s instruction and incubated at 37 °C for 21 to 24 h and was then centrifuged for 15 min at 3500× *g* to harvest the plasma. This test is based on the in vitro stimulation of CD4^+^ and CD8^+^ lymphocytes in whole heparinized blood with a combination of specific SARS-CoV-2 antigens covering the S protein, followed by measurement in the plasma of IFN-ɣ production by ELISA. This assay consisted of the Ag1 tube containing CD4^+^ epitopes derived from the S1 subunit (RBD) of SARS-CoV-2, and the Ag2 tube containing CD4^+^ and CD8^+^ epitopes from S1 and S2 of SARS-CoV-2. The plasma fraction was used to determine the concentration of IFN-ɣ (IU/mL) using QuantiFERON^®^ ELISA and measured the absorbance at 450 nm. The IFN-ɣ standard curve was calculated by using the QuantiFERON R&D Analysis Software. The seropositivity rate was calculated using IFN-ɣ levels from the stimulated tube minus the negative control tube. An IFN-ɣ level of ≥0.15 IU/mL and ≥25% of nil were defined as a positive response against SARS-CoV-2.

### 2.5. Statistical Analysis

G*power software version 3.1.9.6 was used for calculating the sample size. The GraphPad Prism version 7.0 for Microsoft Windows was used for performing the graphical presentation and statistical analyses. The chi-square test and Welch’s ANOVA were performed for categorical analyses of age and sex. Antibody responses against SARS-CoV-2 were designated as geometric mean titers (GMT) with a 95% confidence interval (CI). Neutralization activities and SARS-CoV-2-stimulating IFN-ɣ were presented as medians with interquartile ranges. The Kruskal–Wallis test or Wilcoxon signed rank test (nonparametric) multiple comparison adjustments were used for calculating the differences in antibody titers, percentage inhibition, and IU/mL minus nil between groups. A *p*-value < 0.05 was considered to reveal statistical significance.

## 3. Results

### 3.1. Antibody Responses against SARS-CoV-2 in the BBIBP-CorV-Primed Group

The BBIBP-CorV-primed group consisted of 21 male participants and 30 female participants. The average and median ages were 37.7 and 35.0, respectively. Ten participants were lost to follow-up during the study. The enrollment flow diagram is shown in Figure 1.

Total levels of RBD Ig and anti-RBD IgG at 30 days after two-dose vaccination in the BBIBP-CorV-primed group showed seropositivity rates of 97.8% and 93.6%, respectively. The maximum level of total RBD Ig and anti-RBD IgG was detected in all participants after completing two doses of BBIBP-CorV at 30 days, with GMT 42.8 U/mL and 50.0 BAU/mL, respectively. The waning of antibody titers was observed after 90 days of completed vaccination, as shown in Figure 2 and Appendix A.

### 3.2. Antibody Responses against SARS-CoV-2 in the Third Dose with a Viral Vector or mRNA COVID-19 Vaccine Group

In the third-booster group, healthy participants who received two doses of BBIBP-CorV over 6 ± 1 months were separated into three groups to receive the third dose with a viral vector or mRNA COVID-19 vaccine depending on vaccine availability. The enrolled participants had no underlying disease or had well-controlled disease, were not immunocompromised, or were receiving treatment with ongoing immunosuppressive therapy. The demographic characteristics of all participants, namely those who received vaccines AZD1222 (*n* = 33, median age = 45.0 years), BNT162b2 (*n* = 56, median age = 40.0 years), and mRNA-1273 (*n* = 55, median age = 46.0 years) vaccines; the average age of the participants enrolled in each group was 42.27, 41.89, and 44.02 years, respectively. There were no significant differences in sex and age in each group. The enrollment flow diagram is shown in Figure 1.

The most common symptom induced by the third booster dose of the COVID-19 vaccine in three groups was injection-site pain. The requested injection site and systemic adverse reactions were reported. The most common adverse effects at the injection site and systemic adverse reactions were myalgias, headache, and chilling. The incidence of total local and systemic symptoms in the AZD1222, BNT162b2, and mRNA-1273 groups are shown in Figure 3 and Appendix A. The third dose of AZD1222 had a significantly lower incidence of swelling than BNT162b2 and the mRNA-1273 vaccine did. Fever symptoms had a higher incidence, but the remaining differences were not significant. Any adverse reactions resolved within a few days. After vaccination, no severe adverse events (SAEs) were observed in all third-booster subgroups.

The antibody level before the booster in individuals from all third-booster subgroups showed that the total RBD Ig and anti-RBD IgG levels at baseline were maintained at a low level, similar to those in the BBIBP-CorV-primed group. After receiving the third dose, there was a substantial increase in the highest titers on day 14 and a slight decrease up to day 90, as shown in Figure 4 and Appendix A. Compared with the third-booster group, AZD1222 achieved total RBD Ig and anti-RBD IgG levels that were significantly lower than those achieved with BNT162b2 or mRNA-1273.

### 3.3. Surrogate Virus Neutralization Test

The functionally binding antibody that may inhibit SARS-CoV-2 delta and omicron variants was evaluated by sVNT. According to Figure 5 and Appendix A, before the third dose vaccination, the median percentage of inhibition of participants who received BBIBP-CorV was less than 30.0% for the delta and omicron variants. In the comparison of neutralizing activities between the different third-dose platforms, the highest percentage of inhibition against delta and omicron variants was achieved after the booster on day 14. The median of neutralizing activities against the delta variant showed a 97.0% inhibition in the vaccination with the three vaccines studied. The median neutralizing activities against the omicron variant of the third dose with AZD1222, BNT162b2, and mRNA-1273 were 49.8%, 67.4%, and 53.7%, respectively. On day 28, the percentage of inhibition against omicron variants decreased slightly to less than 30.0% on day 90, while the percentage of inhibition against delta variants was still higher than 90%. The neutralizing capacity of sera against the omicron variant in all third-dose groups was lower than that of the delta variant. In addition, recipients of the third dose of the AZD1222 vaccine had no significant difference in neutralizing activity against the delta variant compared with that of BNT162b2 and mRNA-1273 vaccine recipients but had significantly lower neutralizing activity against the omicron variant (*p*-value < 0.01).

A correlation was plotted between the percentage of inhibition against the SARS-CoV-2 variants, total-RBD Ig, and anti-RBD IgG. Nonlinear regression, a one-phase decay model, could predict immunogenicity titers to neutralize inhibition against delta and omicron variants in the third dose with the different platforms (Appendix A).

### 3.4. SARS-CoV-2 Stimulating IFN-ɣ CD4+/IFN-ɣ CD4+ and CD8+ T-Cell Response

The IFN-ɣ responses to SARS-CoV-2 IFN-ɣ CD4+/IFN-ɣ CD4+ and CD8+ were evaluated, as shown in Figure 6 and Appendix A. Interestingly, potential antiviral-induced IFN-ɣ stimulation of T-cells was notably observed. The marked individual difference in terms of IFN-ɣ release induced by IFN-ɣ CD4+/IFN-ɣ CD4+ and CD8+ ranged from 0.0 to 10.0 IU/mL. In terms of the serological response, the seropositivity rates for the IFN-ɣ CD4+/IFN-ɣ CD4+ and CD8+ levels observed in most participants after the booster dose 14 days were 67.9%/82.1% for AZD1222, 83.3%/93.3% for BNT162b2, and 86.2%/93.1% for mRNA-1273. The production of IFN-ɣ by T-cells was decreased on days 28 and 90, consistent with the total RBD Ig and anti-RBD IgG levels. All booster vaccines showed that humoral immunity also developed cellular immunity. In contrast, both BNT162b2 and mRNA-1273 vaccines could stimulate robust antigen-specific T-cell responses after 14 days with especially high levels of IFN-ɣ. There was a good correlation between the results for IFN-ɣ CD4+/IFN-ɣ CD4^+^ and CD8^+^ (Spearman correlation coefficient, *p*-value < 0.01).

## 4. Discussion

The BBIBP-CorV vaccine is a highly valuable tool against COVID-19 due to its less stringent storage and shipment requirements, which differ from those of mRNA-based vaccines. The BBIBP-CorV vaccine has been approved for use in more than 45 countries around the world, and the interim results of the phase 3 trials indicated 78.1% efficacy in preventing the ancestral variant [28]. Our data showed that immunogenicity could be detected after two doses of the BBIBP-CorV vaccine at 30 days, and the seropositivity rates of total RBD Ig and anti-RBD IgG were 97.8% and 93.6%, respectively. The magnitudes of the total RBD Ig and anti-RBD IgG responses were the highest (GMT 42.8 U/mL, and 50.0 BAU/mL, respectively) in individuals 30 days after having completed two doses of the BBIBP-CorV vaccine and slightly decreased and maintained at a low level until 90 days at levels similar to those as the baseline of the third-dose group which had over 6 ± 1 months of two doses of the BBIBP-CorV vaccine. These results are in line with a previous study reporting that on day 18 after vaccination with the BBIBP-CorV vaccine, more than half of the participants showed negative results for the IgG anti-spike protein antibody test and indicates that the BBIBP-CorV vaccine does not produce a sufficient immune response in most vaccinated individuals [6].

With regard to the side effects of the third dose, previous studies [29,30] have shown that the most reported adverse effects of AZD1222 are pain and tenderness at the injection site, fatigue, and headache. Side effects started on the first day after vaccination and lasted one to three days. The most common side effects for BNT162b2 and mRNA-1273 were pain at the injection site, headaches, fever, flu-like symptoms, and fatigue. Although post-vaccination side effects are the same for all vaccines, the number and severity of these side effects differ significantly according to the type of vaccine. In this study, an incidence rate of injection site and systemic reactions was reported within seven days after boost vaccination, all adverse symptoms were mild in severity and mostly transient. The results suggest that a third booster vaccination in healthy adults aged over 18 years is safe, and these results may help support booster vaccination strategies to be administered in the future [31].

Studies of the third dose of COVID-19 vaccination reported the effectiveness of reducing risk in symptomatic infection and hospitalization rates compared to two doses of vaccination [15,19,22]. Therefore, for the general population, vaccination is necessary. Furthermore, heterologous booster doses (two prime doses with inactive or viral vector vaccine and a third dose with an mRNA vaccine) appear to induce higher levels of immune response than homologous booster doses do, suggesting that heterologous immunization could be considered an alternative to homologous booster doses for immunization programs [22,32,33]. To assess the immunogenicity of a booster vaccination, our study found that a third booster dose with heterologous booster immunization of AZD1222, BNT162b2, and mRNA-1273 showed a high titer of total RBD Ig and anti-RBD IgG compared to that before the third booster. Immunogenicity and neutralizing antibody titers increased significantly on day 14. After 28 days from the third booster dose, the immunogenicity, and neutralizing titers continuously decreased until day 90. In correlation with previous studies, the vaccine effectiveness against delta and omicron variants of a third booster shot of CoronaVac, AZD1222, and BNT162b2 at 14 days after previous priming vaccination with two doses of CoronaVac was 56–80%, 56–90%, and 56–93%, respectively [14,34]. Neutralizing antibodies are the most important parameter for measuring the immune response to prevent disease and reinfections. Although neutralizing antibodies are difficult to investigate in routine laboratories, the level of anti-SARS-CoV-2 IgG is indicative. Previous studies found that neutralizing antibodies and anti-SARS-CoV-2 IgG titers are compatible. The level of total anti-RBD antibodies was strongly correlated with the neutralizing antibodies’ titers of SARS-CoV-2, especially against the ancestral strain [35,36]. This study showed that the titers of total RBD Ig and anti-RBD IgG were correlated with the neutralizing antibody titers against delta and omicron variants. Such results advocate the idea that a higher immune response would provide better neutralizing antibodies in a real-life situation.

For the IFN-ɣ response, previous studies have reported that the third dose of an inactivated SARS-CoV-2 vaccine could generate SARS-CoV-2-specific CD4^+^ and CD8^+^ T-cells and elicit B-cell responses by antibody evolution [37]. This study observed seropositivity rates for the IFN-ɣ CD4+/IFN-ɣ CD4^+^ and CD8^+^ levels in most participants after the booster dose of 14 days and a significant decrease in IFN-ɣ production T-cells after boosting the dose for 14 days, limiting the efficacy of the vaccines. This result is similar to that of a previous report [37] and suggests that a second dose improves an effective antibody response, and further boosters may be needed to retain vaccine effectiveness in adults. Whether enhanced B- and T-cell responses could lead to better protection is one of the key issues concerning boosting strategies. As a result, a third-dose vaccination after six months, even a viral vector or an mRNA vaccine, is in the new guidelines that make it mandatory to get a booster. However, more information is needed for use in a larger population in the future.

The limitations of the current study include the limit of detection of the sVNT assay. Most individuals receiving a booster dose achieved an elevation in antibody levels that was greater than the upper limit of detection. This method may not reflect the actual neutralizing antibody level. Furthermore, the participants lost to follow-up due to breakthrough infection or lost contact during the follow-up period may not reflect the actual values at each time point.

In conclusion, a third-dose heterologous regimen, two initial doses of BBIBP-CorV vaccines followed by a viral vector or mRNA vaccine, induced a robust immunological response. This study found that a two-dose schedule generated good immune memory. However, immunogenicity titers decreased to near or below the lower limit of the seropositivity rate over time. A third dose given 6 ± 1 months after the second dose of BBIBP-CorV was highly effective in recalling a specific SARS-CoV-2 immune response, leading to a significant rebound in antibody levels. This approach would make it possible to optimize the third available booster dose of vaccines without increasing the risk of infection and allowing a faster expansion of vaccination coverage.

## Figures and Tables

**Figure 1 vaccines-10-01071-f001:**
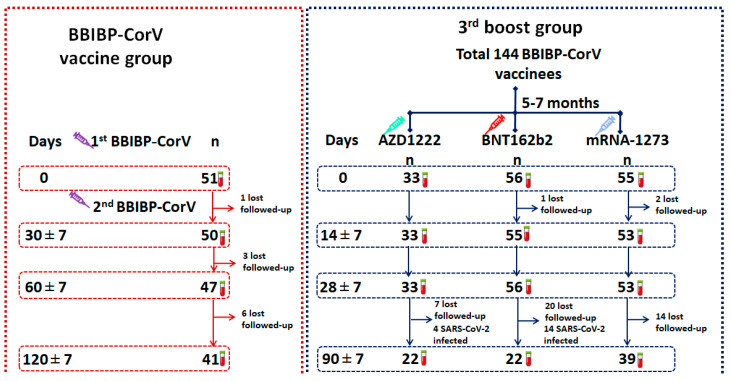
Study diagram for the enrollment of participants in this study. Schematic depicting a total of 51 participants in the BBIBP-CorV-primed group and 144 participants in the third-booster group.

**Figure 2 vaccines-10-01071-f002:**
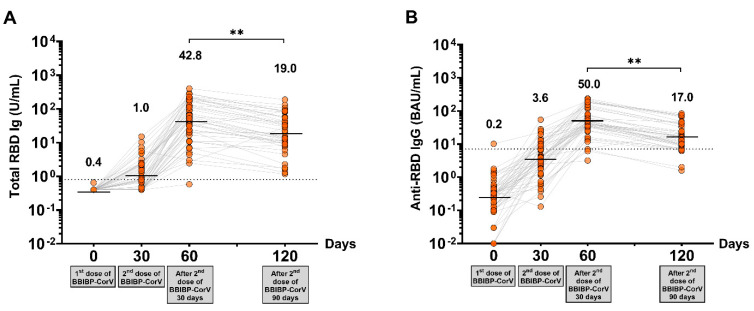
Immune responses against SARS-CoV-2 in the BBIBP-CorV-primed group. (**A**) The total circulating RBD Ig of SARS-CoV-2 (U/mL), and (**B**) circulating anti-RBD IgG of SARS-CoV-2 (BAU/mL). The lines represent GMTs. The dotted lines represent the cutoff titers; *p* < 0.01 (**).

**Figure 3 vaccines-10-01071-f003:**
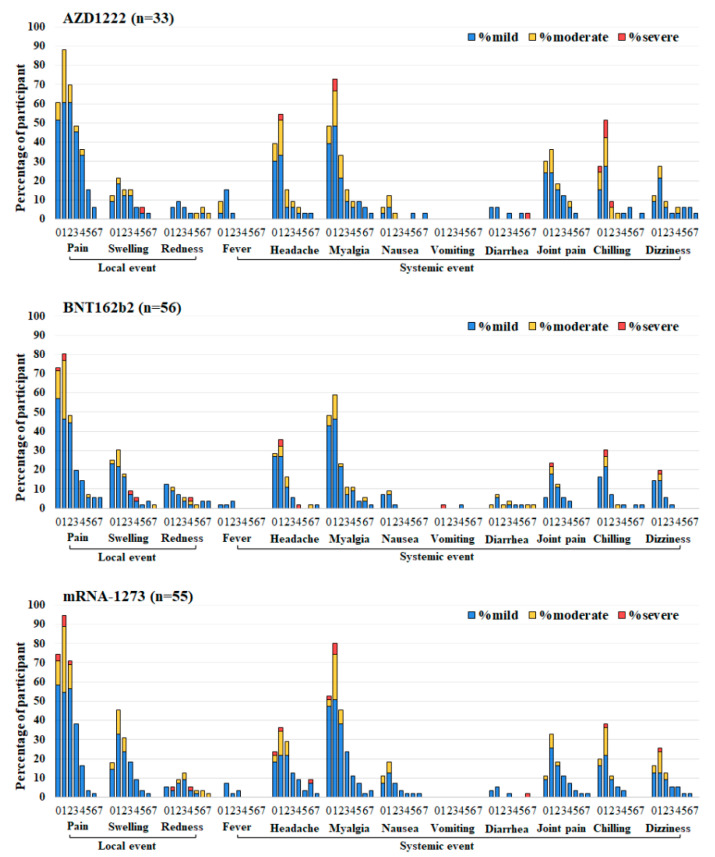
Reactogenicity of a third booster dose of SARS-CoV-2 vaccines at 0–7 days after vaccination. Third-booster subgroups: AZD1222, BNT162b2, and mRNA-1273. The percentages of participants who recorded local and systemic reactions are shown on the Y-axis. Fever was categorized into three levels: mild (38.0 to 38.5 °C), moderate (38.5 to 39.0 °C), and severe (≥39.0 °C). For local and systemic reactions, the grading was classified as mild (easily tolerated with no limitation on normal activity), moderate (some limitation of daily activity), and severe (unable to perform the normal daily activity).

**Figure 4 vaccines-10-01071-f004:**
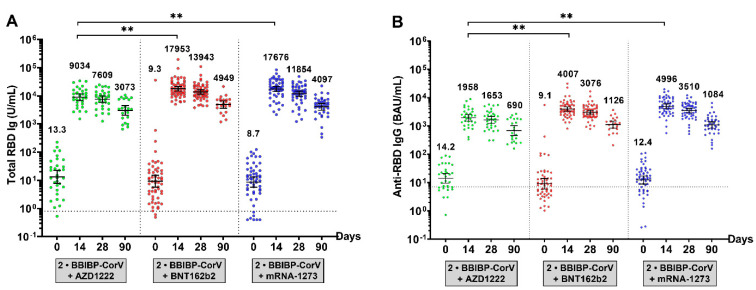
Antibody responses to the SARS-CoV-2 assay in a third-booster group. (**A**) The total RBD Ig circulating of SARS-CoV-2 (U/mL) and (**B**) anti-RBD IgG circulating of SARS-CoV-2 (BAU/mL). Serum samples were obtained from participants who received the third dose with AZD1222 (green), BNT162b2 (red), or mRNA-1273 (blue). The lines represent GMTs (95% CI). The dotted lines represent the cutoff titers; *p* < 0.01 (**).

**Figure 5 vaccines-10-01071-f005:**
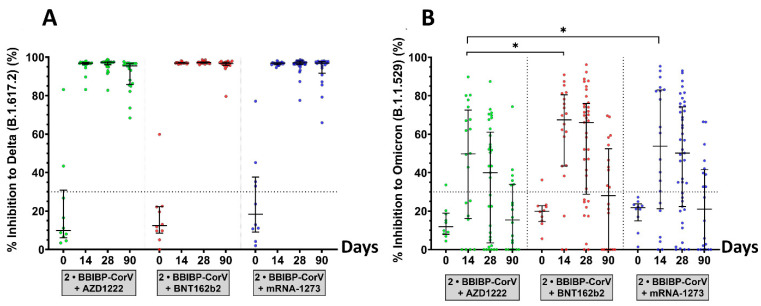
Neutralization activities against SARS-CoV-2 variants B.1.617.2 (delta) and B.1.1.529 (omicron) measured by the surrogate virus neutralization test (sVNT). The subset of serum samples was obtained from a third-booster group. The participants who received the third dose of AZD1222 (green), BNT162b2 (red), or mRNA-1273 (blue) were selected to test sVNT at baseline, 14, 28, and 90 days after the booster dose. (**A**) Neutralizing activity against delta variants and (**B**) neutralizing activity against omicron variants. The lines represent medians with interquartile ranges (IQR). The dotted lines represent the cutoff titers; *p*-value < 0.05 (*).

**Figure 6 vaccines-10-01071-f006:**
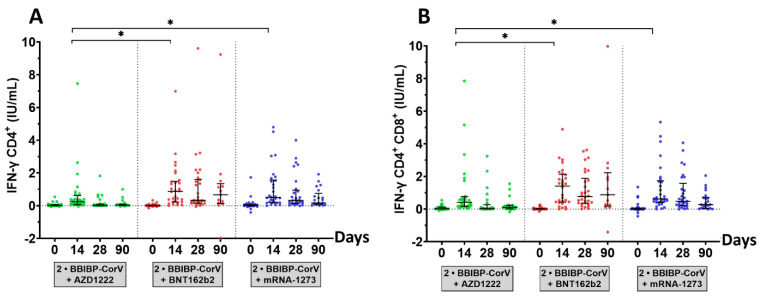
SARS-CoV-2-stimulating IFN-ɣ assay. The subset of heparinized samples was obtained from a third-booster group. Participants who received the third dose of AZD1222 (green), the BNT162b2 (red), or the mRNA-1273 (blue) were stimulated by a CD4+ epitope derived from RBD (Ag1) and a CD4+/CD8+ epitope derived from S1 and S2 subunits (Ag2) and incubated in a QFN blood collection tube for at least 21 h. QFN IFN-ɣ ELISA evaluated the plasma fraction on days 0, 14, 28, and 90. (**A**) The IFN-ɣ produced by CD4+ T-cells and (**B**) the IFN-ɣ produced by CD4+ and CD8+ T-cells. The lines represent medians with interquartile ranges (IQR). The dotted lines represent cutoff titers. Statistical analysis was performed using the Wilcoxon signed-rank test (two-tailed); *p*-value < 0.05 (*).

## Data Availability

The authors confirm that the data supporting the findings of this study are available within the article.

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
