# Peer review of "Immunogenicity Following Two Doses of the BBIBP-CorV Vaccine and a Third Booster Dose with a Viral Vector and mRNA COVID-19 Vaccines against Delta and Omicron Variants in Prime Immunized Adults with Two Doses of the BBIBP-CorV Vaccine"

_vaccines, 2022, doi:10.3390/vaccines10071071_

Round 1

Reviewer 1 Report

Dear authors, 

Overall work here, I believe the authors have done an exemplary job in preparing this manuscript. The level of scientific rigor is apparent, and the attention to detail in regard to every aspect of the replication is appreciated. 

I have a few minor suggestions that the authors might consider, but all of them would be moving forward. 

Please check the manuscript for typo errors and English corrections.

Author Response

Response comment to reviewer 1

Comment: Overall work here, I believe the authors have done an exemplary job in preparing this manuscript. The level of scientific rigor is apparent, and the attention to detail in regard to every aspect of the replication is appreciated. 

I have a few minor suggestions that the authors might consider, but all of them would be moving forward. 

Please check the manuscript for typo errors and English corrections.

Response: We would like to sincerely thank the reviewer for the very encouraging comments.  We have responded to revise some grammatical or typographical errors as follows.

Reviewer 2 Report

I want to congratulate the authors of this paper for the good scientific content and the perfect presentation of the results.  This study aimed to evaluate antibody responses in individuals vaccinated with two doses of BBIBP-CorV vaccine and to explore the boosting effect of the different vaccine platforms in BBIBP-CorV-primed healthy adults, including viral vector vaccine and mRNA vaccines. The results showed that, in the BBIBP-CorV prime group, the total receptor-binding domain (RBD) immunoglobulin (Ig) and anti-RBD IgG levels waned significantly at 3 months after receiving the second dose. 

Author Response

Response comment to reviewer 2

Comment: I want to congratulate the authors of this paper for the good scientific content and the perfect presentation of the results.  This study aimed to evaluate antibody responses in individuals vaccinated with two doses of BBIBP-CorV vaccine and to explore the boosting effect of the different vaccine platforms in BBIBP-CorV-primed healthy adults, including viral vector vaccine and mRNA vaccines. The results showed that, in the BBIBP-CorV prime group, the total receptor-binding domain (RBD) immunoglobulin (Ig) and anti-RBD IgG levels waned significantly at 3 months after receiving the second dose. 

Response: We would like to sincerely thank the reviewer for the very encouraging comments.

Reviewer 3 Report

This work is interesting and the manuscript is well prepared.

However, micrograph of the tested cells/tissues should be added which would help to better understand the real scenario on cellular level. If possible, the cell samples should be collected from the participants for testing.

Author Response

Response comment to reviewer 3

Comment: This work is interesting and the manuscript is well prepared.

However, micrograph of the tested cells/tissues should be added which would help to better understand the real scenario on cellular level. If possible, the cell samples should be collected from the participants for testing.

Response: We would like to thank the reviewer for the comments to strengthen the information provided by this study. In this study, we collected the whole blood sample and then activated it in QuantiFERON SARS-CoV-2 antigen kit. After activation, we collected only plasma samples (The activated cells were removed). However, we appreciate the suggestion. It will be useful for further experiment design.

Reviewer 4 Report

Thanks for the opportunity to review the manuscript titled "Immunogenicity following two doses of BBIBP-CorV vaccine and a third booster dose with viral vector and mRNA COVID-19 vaccines against delta and omicron variants in prime immunized adults with two doses of BBIBP-CorV vaccine". It is a well written and organized manuscript about the determination of antibodies of the third dose of COVID-19 vaccination using a viral vector and mRNA vaccines. The research topic is appealing and actual, it explains several interesting points; figures and tables are explicative and overall is well written. I suggest publishing it with some minor changes.

1.    Why is the last analysis of the BBIBP-CorV group done at 120 days and the last analysis of the booster group at 90 days?

2.    Authors do not mention in the text the 16 patients lost in the 3rd boost group.

Author Response

Response comment to reviewer 4

Comment: Thanks for the opportunity to review the manuscript titled "Immunogenicity following two doses of BBIBP-CorV vaccine and a third booster dose with viral vector and mRNA COVID-19 vaccines against delta and omicron variants in prime immunized adults with two doses of BBIBP-CorV vaccine". It is a well written and organized manuscript about the determination of antibodies of the third dose of COVID-19 vaccination using a viral vector and mRNA vaccines. The research topic is appealing and actual, it explains several interesting points; figures and tables are explicative and overall is well written. I suggest publishing it with some minor changes.

  1. Why is the last analysis of the BBIBP-CorV group done at 120 days and the last analysis of the booster group at 90 days?

Response: We apologize for the confusion. The BBIBP-CorV vaccine is recommended that all vaccinated individuals receive two doses at an interval of 3-4 weeks between the first and second dose. Therefore, for the BBIBP-CorV primed group, 120 days is the 90 days after receiving a complete dose (2nd dose) of BBIBP-CorV.

  1. Authors do not mention in the text the 16 patients lost in the 3rd boost group.

Response: We thank the reviewer for the comments. We added the information of lost follow-up participants in the limitation in lines 394-399.